# Effects of Different Gluten Proteins on Starch’s Structural and Physicochemical Properties during Heating and Their Molecular Interactions

**DOI:** 10.3390/ijms23158523

**Published:** 2022-07-31

**Authors:** Tao Yang, Pei Wang, Qin Zhou, Yingxin Zhong, Xiao Wang, Jian Cai, Mei Huang, Dong Jiang

**Affiliations:** College of Agriculture, Nanjing Agricultural University, No. 1 Weigang Road, Nanjing 210095, China; 2017101046@njau.edu.cn (T.Y.); wangpei@njau.edu.cn (P.W.); qinzhou@njau.edu.cn (Q.Z.); yingxinzhong@njau.edu.cn (Y.Z.); xiaowang@njau.edu.cn (X.W.); meihuang@njau.edu.cn (M.H.); jiangd@njau.edu.cn (D.J.)

**Keywords:** gluten, starch, heat, interaction, wheat

## Abstract

Starch–gluten interactions are affected by biopolymer type and processing. However, the differentiation mechanisms for gluten–starch interactions during heating have not been illuminated. The effects of glutens from two different wheat flours (a weak-gluten (Yangmai 22, Y22) and a medium-strong gluten (Yangmai 16, Y16)) on starch’s (S) structural and physicochemical properties during heating and their molecular interactions were investigated in this study. The results showed that gluten hindered the gelatinization and swelling of starch during heating when temperature was below 75 °C, due to competitive hydration and physical barriers of glutens, especially in Y22. Thus, over-heating caused the long-range molecular order and amylopectin branches of starch to be better preserved in the Y22-starch mixture (Y22-S) than in the Y16-starch mixture (Y16-S). Meanwhile, the starch’s degradation pattern during heating in turn influenced the polymerization of both glutens. During heating, residual amylopectin branching points restricted the aggregation and cross-linking of gluten proteins due to steric hindrance. More intense interaction between Y16 and starch during heating mitigated the steric hindrance in starch–gluten networks, which was due to more residual short-range ordered starch and hydrogen bonds involved in the formation of starch–gluten networks in Y16-S during heating.

## 1. Introduction

Starch and gluten are the most important macromolecules in wheat flour, which greatly affect the functional and processing characteristics of wheat flour [1]. Wheat flour is one of the most consumed materials of cereal products and needs a lot of processing procedures such as mixing, fermentation, heating, and so on to achieve the ideal product [2]. Heat processing is an important unit operation in the manufacture of cereal food products, which can cook the flour or dough to form products with special flavor and texture [3]. Food properties generally depend on its chemical composition, structure, and temperature during heating. One challenge to industrial users is the limited value of available data caused by insufficient reporting on composition, temperature, and measurement errors, among others. In addition, inadequate thermophysical and chemical data exist for both high- and low-temperature conditions, while these data are highly useful for preliminary design, heat transfer calculations, food production, and food quality assessment. Therefore, a systematic and in-depth understanding of behaviors and interactions of starch and gluten during heating processing contributes to controlling and regulating the products’ quality.

The properties of starch and gluten usually change with temperature. When processed under different heating temperature, protein configuration and molecular weight changed, which are driven by the release and rearrangement of chemical forces (disulfide bond, hydrophobic interactions, hydrogen bond, ionic bonds, and so on) [4]. Gelatinization is the most significant change in wheat starch during heating, which is characterized by granule swelling, amylose leaching, long-range molecular order disruption, and formation of a three-dimensional starch network [5]. Furthermore, in models using purified starch and proteins from meat [6] as well as soy [7], interactions among these macromolecules have been shown. The proteins form the gluten network and wrap starch granules, hence, starch gelatinization is influenced by the barrier effect of gluten and competitive hydration effect between gluten and starch [3,8]. The starch–gluten interactions are of specific importance during heating because of heat-induced changes within these two macromolecules. However, since previous studies rarely focused on interactions between starch and gluten during heating, little is known about the specific interaction mechanism in such model systems, for example, how residual short-range order in gelatinized starch influences gluten polymerization.

The interactions between macromolecules under the action of heat are complex. For example, un-gelatinized starch granules may affect the gluten and interact with gluten, differing with granule size [9]. Gelatinized starch gels may compete with the reaction conditions of starch–gluten matrix formation and then affect the stabilization of gluten networks [10]. Monomeric proteins such as gliadin, albumin, and globulin have intense interaction with amylose in gelatinized starch, while they have strong interactions with amylopectin in retrograde starch [11]. In addition, the effects of gluten on the starch gelatinization is different. For example, in the case of pasting viscosity, the effect of gliadin is greater than that of glutenin [12]. A study theoretically reported that starch can enhance the hydrophobic interaction of glutenin aggregates [13]. Meanwhile, the physicochemical properties of starch during heating is based on the increased hydration of the starch granules and its intra- and intermolecular hydrogen bonds [14]. Therefore, with a change in temperature, hydrophobic interaction and hydrogen bond can affect the interaction between starch and protein, but the specific situation still remains unclear.

However, the presence of lipids in real cereal-based food systems may affect the physicochemical behaviors of starch and gluten during processing through the potential formation of starch–lipid or starch–lipid–protein complexes, causing great difficulties in investigating the starch–gluten interaction mechanism. Use of purified starch and gluten as a model material has led to a deep and comprehensive understanding of the interaction mechanisms of starch granules and the gluten network during thermal food processing. In this study, gluten from two wheat varieties with the same high-molecular weight (HMW) glutenin profile but of different strength was used to probe the transformation and interaction between the transient and continuous states of starch and gluten during a gradual heating process. This study may help us better understand the structure–function relationship of gluten–starch during food processing, thereby relating the thermal processing of starch and gluten to further provide guidance for the manufacturing of wheat flour products.

## 2. Results

### 2.1. Pasting and Thermal Properties

The pasting parameters of native starch and pre-heated mixtures are shown in Table 1; native starch had the highest peak and final viscosities, and the viscosities in both pre-heated mixtures increased with the temperature from 25 to 50 °C, while it sharply decreased from 50 to 75 °C, while there was a slight increase from 75 to 100 °C. Y22 and Y16 had different effects on the starch viscosity properties during heating, where peak viscosity increased by 14.29% and 2.98% for pre-heated Y22-S and Y16-S mixtures at 50 °C compared with 25 °C, respectively. The peak viscosity sharply decreased in two mixtures when pre-heated temperature increased from 50 to 75 °C, and the lower peak viscosity was observed in pre-heated Y16-S mixture. In addition, other RVA parameters (trough viscosity, breakdown viscosity, final viscosity, and setback) showed the similar change trend in Y22-S and Y16-S mixtures during heating.

Thermal properties of native starch and pre-heated mixtures are summarized in Table 1, which features the gelatinization temperature and enthalpy as the main parameters of DSC. To, Tp, and Tc of native starch were 55.30, 61.38, and 65.71 °C, respectively. Y22 and Y16 had different effects on starch gelatinization temperature during heating, where the To, Tp, and Tc increased significantly by the increment of 2.71%, 1.17%, and 2.88% in pre-heated Y22-S mixture and 4.70%, 4.76%, and 4.25% in pre-heated Y16-S mixture at 25 °C compared with native starch, respectively. With the increase in temperature, To, Tp, and Tc in both two pre-heated mixtures significantly increased, especially obviously in pre-heated Y16-S mixture. The ΔH of native starch was 6.3 J/g, and the ΔH was significantly decreased by 26.34% and 31.38% in pre-heated Y22-S and Y16-S mixtures at 25 °C compared with native starch, respectively. Meanwhile, the ΔH decreased obviously in both two mixtures as the pre-heated temperature increased, especially from 50 to 75 °C. No gelatinization peaks were detected for samples that were heated at 100 °C.

### 2.2. Solubility and Swelling Power

As shown in Table 1, the native starch had the highest solubility (25.03%) and swelling power (27.40 g/g). The starch’s solubility and swelling power both decreased significantly in pre-heated Y22-S and Y16-S mixtures at 25 °C compared with native starch. The starch’s solubility and swelling power significantly increased both in pre-heated Y22-S and Y16-S mixture at 50 °C compared with 25 °C, while significantly decreasing at 75 °C, which decreased by 27.44% and 52.88% for swelling power and 24.99% and 23.55% for solubility compared with 50 °C, respectively.

### 2.3. Crystalline and Short-Range Ordered Structure

The XRD patterns and crystallinity of starch during heating are illustrated in Figure 1. Apparently, the native starch, Y22-S and Y16-S mixtures pre-heated at 25 and 50 °C showed typical A-type diffraction pattern, with two singlet peaks at 2θ-15° and 2θ-23°, a doublet peak at 2θ-17° and 2θ-18°, and a weak peak at 2θ-20°. The Y22-S (35.39%) and Y16-S (37.55%) mixtures pre-heated at 25 °C had higher starch crystallinity than native starch (33.83%). The starches in both mixtures heated at 75 and 100 °C were mostly amorphous-structured but also contained V-type crystals, as shown in the peaks at 20° (2θ), suggesting that the tiny quantity of lipid (0.45%) in native starch contributed to the creation of a lipid–starch complex at higher temperatures. Even though the Y22-S and Y16-S mixtures remained in the crystalline pattern of starch granules, as shown in Figure 1, the starch crystallinity of Y22-S and Y16-S mixtures was significantly decreased from 35.39% to 24.93% and 37.55% to 24.58% from 25 to 50 °C, respectively. Meanwhile, the starch crystallinity in the Y22-S mixture heated at 75 °C was higher than that in the Y16-S mixture. There was a small difference between the Y22-S mixture (1.85%) and Y16-S mixture (1.28%) heated at 100 °C.

FT-IR and FT-Raman spectroscopy were performed to provide information on the short-range order of starch. The absorbance ratio of 1047/1022 cm^−1^ is indicative of the degree of order in the outer regions of starch while the ratio of 1022/995 cm^−1^ denotes proportions of amorphous versus ordered structures. As shown in Table 2, native starch had a significantly higher ratio of 1047/1022 cm^−1^ and significantly lower ratio of 1022/995 cm^−1^ than all Y22-S samples. In these samples, the 1047/1022 cm^−1^ and 1022/995 cm^−1^ ratios progressively decreased and increased as soon as samples were heated. In contrast, the ratios were only significantly different between Y16-S and native starch after samples had been heated to 50 °C (1047/1022 cm^−1^) or 75 °C (1022/995 cm^−1^). Moreover, while the 1047/1022 cm^−1^ ratio decreased and the 1022/995 cm^−1^ ratio also increased in Y16-S over heating, the ratios were, at all temperatures, higher and lower, respectively, than in Y22-S samples.

Although some vibrations may give rise to signals in both the Raman and FTIR regions of the electromagnetic spectrum, FTIR spectroscopy is best for asymmetrical vibrations of polar groups, whereas Raman spectroscopy is best at detecting symmetrical internal vibrations of nonpolar groups. The full width at half maximum (FWHM) of the Raman band at 480 cm^−1^ was calculated to characterize the short-range ordered structures in starch, and higher FWHM values at 480 cm^−1^ represent a lower proportion of short-range ordered structures. As shown in Table 2, the FWHM at 480 cm^−1^ decreased when the temperature increased from 25 to 50 °C, and then increased with the further increasing in temperature in both mixtures. The FWHM at 480 cm^−1^ increased from 16.31 to 17.74 in the Y22-S mixture and from 16.14 to 17.54 in the Y16-S mixture with the increase of temperature from 25 to 100 °C, respectively. Additionally, the IR ratio of 1022/995 cm^−1^ significantly increased compared with native starch in both mixtures with the increase in temperature, while there was no significant difference in the IR ratio of 1022/995 cm^−1^ for the Y16-S mixture between 75 and 100 °C.

### 2.4. CLSM Characterization

The morphology of starch during heating was observed by CLSM (Figure 2). For the native starch and mixtures of Y22-S and Y16-S at 25 and 50 °C, many complete round or oval starch crystal granules were observed. With the increase in temperature, starch granules gradually swelled (especially at 75 °C) and disintegrated. The presence of starch residues blurred the boundaries of the starch granules, where the Y16-S mixture had more broken and swollen starch granules compared with the Y22-S mixture. Meanwhile, spongy structures of starch gels in both mixtures began to develop at 75 °C, and it was more obvious in the Y22-S mixture. At a higher heating treatment of 100 °C, the granule morphology of starch disappeared.

### 2.5. SDS-PAGE Analysis

Non-reducing and reducing SDS-PAGE were performed to analyze the protein polymerization during heating. As shown in non-reducing SDS-PAGE (Figure 3A), the band density of SDS-soluble proteins (130–200 KDa) decreased continuously with the increase in temperature in both Y22 and Y16, and almost disappeared at 100 °C, which was especially more obvious in Y16. To further probe the interaction between starch and gluten in the thermal process, we comparatively analyzed the non-reducing SDS-PAGE (Figure 3B) and reducing SDS-PAGE (Figure 3C) of gluten–starch mixtures during heating. As shown in Figure 3B, the band density of SDS-soluble protein decreased continuously with the increase in temperature in both two mixtures. Although almost disappeared at 100 °C, the band density was higher than that in corresponding gluten samples, and it was more pronounced in the Y22-S mixture. Furthermore, as shown in Figure 3D, the band intensity of the SDS-soluble protein in Y22-S and Y16-S was significantly higher than that in corresponding Y22 and Y16 samples at 100 °C; meanwhile, the intensity of the SDS-soluble protein in Y16 and Y16-S mixtures was significantly lower than that in Y22 and Y22-S samples at 100 °C. As shown in Figure 3C, the bands of SDS-soluble protein disappeared with the addition of β-ME; at the same time, the band density of the high-molecular-weight glutenin subunit (HMW-GS) (75–120 KDa) and α/β-gliadin (LMW-GS) (28–40 KDa) decreased continuously with the increase in temperature, while it changed differently in different mixtures during heating, where the band density of HMW-GS decreased more obviously in the Y16-S mixture than in the Y22-S mixture, as shown in Figure 3E.

### 2.6. Chemical Interactions

The changes in chemical interactions in two mixtures during heating are shown in Table 3. With the increase in temperature, the proportions of different chemical interactions were different: where the ionic and hydrogen bonds were relative lower, hydrophobic interaction and disulfide bond were relative higher. Ionic and hydrogen bonds increased or fluctuated slightly during initial heating from 25 to 50 °C, while they decreased significantly at higher temperature, especially when the temperature exceeded 75 °C. Apart from these two chemical interactions, hydrophobic interaction decreased significantly with the increase in temperature and greatly decreased from 50 to 75 °C, while disulfide bonds increased obviously with the increment of 53.37% and 69.47% for Y22-S and Y16-S mixtures, respectively. The Y16-S mixture had higher contents of hydrogen bond, hydrophobic interaction, and disulfide bond at higher temperature than those in the Y22-S mixture.

### 2.7. Molecular Docking

A molecular docking program was used to probe the in-silico protein–starch interaction. Figure 4A,D show the best docking conformations with the lowest binding energies between the starch fragment with HMW-GSs from Y16 (−8.8 kcal/mol) and Y22 (−8.4 kcal/mol), respectively. The results showed highly similar docking poses and sites between two docking conformations, indicating the similar interaction pattern between HMW-GS and starch. Figure 4B,C,E,F show the main interaction forces between amino acid residues and starch fragment, where the starch glucose unit formed a total of 33 and 30 hydrogen bonds with HMW-GS from Y16 (21 residues) and Y22 (16 residues) and a total of 3 and 2 hydrophobic interactions with HMW-GS from Y16 and Y22.

## 3. Discussion

Previous research hypothesized that glutenin comprises the network and embeds the starch granules which are covered by gliadin [15,16]. This research also reported that the variation in gluten protein influenced the interaction between gluten components and starch molecules. Taking this into account, purified starch and gluten were used to establish the mixture model system in this study to analyze starch–gluten interaction during heating. The pre-heated Y16-S mixture at 75 °C had lower swelling power, PV, and FV compared with the pre-heated Y22-S mixture; this could be interpreted by the intense hydration competition between Y16 and starch at 75 °C, as the starch gelatinization was influenced by water availability. Meanwhile, Y16 had a higher glutenin/gliadin ratio, and it was harder to confine water mobility due to the glutenin’s higher mobility of water molecules [17]. This could explain why the swelling power, solubility, and viscosity of the pre-heated Y16-S mixture at 75 °C decreased more than the pre-heated Y22-S mixture. Moreover, due to stronger hydrophilicity, there are many hydrophilic groups on the surface of gliadin [17]. Y22 with a higher gliadin/glutenin ratio could interact with amylopectin double helices more and resulted in lower degradation of amylopectin during heating. Moreover, the result of a higher ΔH value of pre-heated Y22-S mixture may be due to this account (Table 1). Besides, Y16 with a higher glutenin ratio would have a higher proportion of hydrophobic residues and was related to higher Tp value, as hydrophobic residues could involve in the denaturation mechanism of starch [18], resulting in a higher Tp value of pre-heated Y16-S mixture compared with pre-heated Y22-S mixture. The result of a lower relative crystallinity value in the Y16-S mixture at higher temperature compared with Y22-S indicated that more long-range molecular order structure was disrupted in the Y16-S mixture at higher temperature, and this was also consistent with the above results. Notably, the PV, TV, FV, and SV of gluten–starch mixtures heated at 100 °C increased compared with mixtures heated at 75 °C; this may be attributed to the cold-water swelling formation and was capable of increasing viscosity [19]. Meanwhile, the Y16-S mixture heated at 100 °C had higher viscosities compared with the heated Y22-S mixture, which may be related to the formation of more glutenin macropolymer (GMP) in the Y16-S mixture heated at 100 °C, as could be observed in Figure 3B,D, and these were in line with the results reported by Li et al. [12].

Since proteins can interact with starch molecules, the mixture or addition of protein can affect the structural changes of starches at the molecular level [20]. No gelatinization peaks were detected for gluten–starch mixtures that were heated at 100 °C, indicating complete starch gelatinization under the 100 °C condition; this observation was consistent with the XRD and CLSM results. While the increase in To, Tp, and Tc values in gluten–starch mixtures that were heated at 25, 50, and 75 °C may be related to these heated mixtures, they were likely to remain as more stable starch crystallites after less stable crystallites melted during heating [21]. These results were consistent with data shown in Table 2, which indicated that some degree of short-range order remained over heating. The change in short-range molecular order was influenced by the type of gluten present. At various temperatures, Y16-S had a higher absorbance ratio at 1047/1022 cm^−1^ and lower values for the 1022/995 cm^−1^ ratio as well as FWHM at 480 cm^−1^ than Y22-S, suggesting that more of the short-range ordered structure of starch remained in Y16-S over heating than in Y22-S. Previous works revealed that less change in the structural order of starch is mainly attributed to the close link between starch and other ingredients, i.e., lipid or protein [22,23]. Our data indicated that Y16 had stronger interactions with starch during heating compared to Y22, and this preserved the short-range order in starch to a larger extent. Meanwhile, lower binding energies and more interaction forces in starch–HMW-GS from Y16 docking conformations were detected compared with starch–HMW-GS docking conformations, further indicating the strong interaction between starch and Y16.

With the increase in temperature, chemical interactions changed significantly among gluten proteins, resulting in new surface properties and enlarged protein aggregates [24]. Therefore, the new chemical interaction bonds are the major drivers to promote the self-aggregation and cross-linking of protein. Integrating the result of non-reducing SDS-PAGE (Figure 3A), Y16 formed more glutenin macro polymers (GMP) at higher temperature compared with Y22, as the SDS-soluble protein (130–200 KDa) decreased more in Y16 after heating at 100 °C. Notably, we observed that the density of SDS-soluble protein of gluten–starch mixtures during heating (Figure 3B) was relatively higher than that in corresponding gluten samples, indicating that the starch affected the aggregation and cross-linking during heating. Although it is harder for starch and protein to form the crystalline inclusion complexes due to thermodynamic incompatibility [25], starch could affect the gluten–starch interaction in the gluten–starch network. Meanwhile, less GMP was formed in the Y22-S mixture compared with the Y16-S mixture at higher temperature, as shown in the results of non-reducing SDS-PAGE (Figure 3B) and disulfide bonds (Table 3) of gluten–starch during heating, indicating that starch affected the aggregation and cross-linking of different gluten during heating in starch–gluten networks. Furthermore, Chen et al. [26] reported that the amylopectin promoted the disruption of intra- and inter-molecular interactions of heat-induced aggregates and the unfolding of protein chains. Therefore, the lower degradation of amylopectin resulted in the greater integrity of amylopectin in the Y22-S mixture during heating. Due to its large molecular weight and abundance of branching points, amylopectin was more conducive to disrupting the interactions of heat-induced gluten aggregates and hindering the exposure of hydrophobic groups initially buried in the interior (Table 3), consequently increasing protein chain mobility. Meanwhile, Hirai et al. [27] suggested that more branching points led to more steric hindrance, which hindered protein unfolding and aggregation. Hence, more branching points in the Y22-S mixture were not conducive to GMP formation at higher temperatures compared with the Y16-S mixture. Furthermore, the differences in chemical interactions and binding modes in different model mixtures during heating could be another reason. More chemical interactions (hydrogen bond, hydrophobic interaction, and disulfide bond) in the Y16-S mixture at higher temperature could contribute to the steric hindrance mitigation between starch and Y16 (Table 3). A larger contribution of disulfide bonds was in line with expectations for Y16, and more S-S groups in Y16-S mixture would accelerate the gluten polymerization and improve the stability of protein molecules against external factors (such as steric hindrance) during heating. Though relative minor differences of non-covalent interactions (especially hydrogen bond) were observed between Y22-S and Y16-S at higher temperature, the result of molecular docking (Figure 4) showed that the starch glucose unit formed a total of 33 and 30 hydrogen bonds with single HMW-GS from Y16 (21 residues) and Y22 (16 residues) and a total of 3 and 2 hydrophobic interactions with HMW-GS from Y16 and Y22. These results suggested that more non-covalent interactions contributed to the stability and polymerization of Y16 against the steric hindrance in the Y16-S mixture compared with the Y22-S mixture at higher temperature. The result of reducing SDS-PAGE (Figure 3C,E) shows that the band intensity of HMW-GS in the Y16-S mixture at higher temperature was weaker than that of the Y22-S mixture, suggesting that more HMW-GSs were polymerized in the Y16-S mixture and that the aggregation ability of the Y16-S mixture at higher temperature was stronger than that of the Y22-S mixture. The intense interaction between starch and HMW-GS from Y16 in starch–HMW-GS docking conformation (Figure 4C,F) was also consistent with this result. Moreover, although more long-range molecular order was disrupted in the Y16-S mixture at higher temperature, more short-range molecular order remained in the Y16-S mixture at higher temperature compared with that in the Y22-S mixture (Table 2). Therefore, this result became the indirect evidence for the intense interaction between starch and Y16 during heating, as short-range ordered structures in starches are said to be maintained mainly by hydrogen bonding [28], and proteins can interact with starch molecules mainly through hydrogen bonding [20].

## 4. Materials and Methods

### 4.1. Materials

Commercial wheat starch (starch content, 98.85%, on dry basis) was purchased from Yuanye Biological Co., Ltd., Shanghai, China. The contents of moisture, ash, lipid, protein, amylose, and A- and B-type starch granule were 13.11%, 0.19%, 0.45%, 0.76%, 23.4%, 70.32%, and 29.68%, respectively. Two winter wheat (*Triticum aestivum* L.) cultivars, a weak-gluten wheat Yangmai 22 and a medium-gluten wheat Yangmai 16, were used in this study. Yangmai16 and Yangmai22 are two widely cultivated varieties in the middle and lower reaches of the Yangtze River. They have the same high-molecular-weight glutenins (HMW-GSs) but with different gluten strength. Wheat grains were milled using flour miller (ZS70-II, grain, and oil foodstuff machine factory, Zhuozhou, China) with a 100 μm mesh sieve. The gluten indices from medium and weak gluten wheat flour were 88.60 and 63.00, respectively. The glutenin to gliadin ratio was 1.09:1 for weak-gluten wheat flour and 1.52:1 for medium-gluten wheat flour. The ratio of high-molecular-weight glutenin subunit to low-molecular-weight glutenin subunit (HMW/LMW-GS) was 0.29:1 for weak-gluten wheat and 0.38:1 for medium-gluten wheat flour. The glutens from two different wheat variety flours were isolated and extracted following the method described by Wang et al. [17]. After lyophilizing, the samples were milled for further analysis. The glutens from Yangmai 22 and Yangmai 16 were designated as Y22 and Y16, and these had 79.6% and 80.7% protein content, respectively.

### 4.2. Preparation of Starch–Gluten Mixture Samples with Hydrothermal Treatment

First, 3.0 g (dry basis) of starch and gluten samples were accurately weighed in an aluminum case to prepare the starch suspension via adding with 25 mL of distilled water to prepare starch and gluten suspensions, respectively. The starch suspension was added to 0.50 g (dry basis) of gluten followed by 30 s of homogenization to prepare the starch–gluten mixture, then, the starch suspension, the starch–gluten mixture, and gluten suspension were sealed in aluminum case and allowed to stand for 12 h before heating with Rapid Viscosity Analyzer (RVA) program (RVA-3D super type, Newport Scientific) at 25, 50, 75, and 100 °C with a shear speed of 160 rpm/s for 10 min. The suspensions treated with different temperatures were sampled and stored in liquid nitrogen. After lyophilizing, the samples were ground into powder using a high throughput tissue grinder TL 2020 (DHS Life Science and Technology Co., Ltd., Beijing, China) and passed through an 80-mesh sieve for further analysis. The control of native starch was designated as S. The treatments of Y16 and S mixture at 25, 50, 75, and 100 °C were designated as Y16-S-25 °C/50 °C/75 °C/100 °C, and Y22 and S mixture at 25, 50, 75, and 100 °C were designated as Y22-S-25 °C/50 °C/75 °C/100 °C, respectively. The moisture contents of lyophilized Y16-S and Y22-S mixtures were 10.52% and 10.88%, respectively.

### 4.3. Pasting and Thermal Properties

The pasting property of the prepared sample was analyzed with Rapid Viscosity Analyzer 130 (RVA-3D super type, Newport Scientific), following the method described by Chang et al. [29]. The thermal property of the prepared sample was determined by differential scanning calorimetry 8000 (DSC) (PerkinElmer, Waltham, MA, USA), according the method described by Yuan et al. [30].

### 4.4. Starch Swelling Power and Solubility

Starch swelling power and solubility were determined according the method described by Leach [31]. The pre-heated samples were mixed with distilled water (2%, *w*/*v*) and heated at 100 °C for 45 min with stirring and then immediately cooled. After centrifugation (16,000 rpm, 20 min), the supernatant was discarded and dried at 105 °C to a constant weight. Swelling power (g/g) was calculated as the ratio of the wet weight of precipitated colloid to its dry weight, and solubility (%) was calculated as the ratio of dry weight of water-soluble starch in supernatant to the initial dry weight of starch.

### 4.5. X-ray Diffraction (XRD)

The XRD patterns of the prepared sample were determined by an X-ray diffractometer (TD-3500, Tongda, China), which was operated at 40 KV and 40 mA with Cu Kα radiation (λ = 0.154 nm). The sample was operated with diffraction angle (2θ) ranged at 5°–40° at a speed of 0.02° at room temperature. The relative crystallinity (%) was calculated by software MDI Jade 5.0.

### 4.6. Fourier Transform-Infrared (FT-IR) and Raman (FT-Raman) Spectroscopy

The short-range ordered structure of the starch sample was determined with Thermo Nicolet Nexus FTIR (Thermo Scientific, Waltham, MA, USA), following the method previously described by Yang et al. [32]. The detected bands at 1047, 1022, and 995 cm^−1^ were used to further calculate the IR ratio of 1047/1022 cm^−1^ and 1022/995 cm^−1^. The ratio of absorbance 1022/995 cm^−1^ was used to measure the ratio of amorphous to ordered carbohydrate structure in starch, and the ratio of absorbance 1047/1022 cm^−1^ was used to quantify the ordered degree of starch external region of starch

The short-range ordered structure was also determined by FT-Raman spectra (NEXUS-870, Thermo 150 Nicolet Inc., Waltham, MA, USA), as reported by Brandt et al. [33]. The full width at half maximum (FWHM) of the Raman band at 480 cm^−1^ was obtained using the WIRE 2.0 software, and higher FWHM values at 480 cm^−1^ represent a lower proportion of short-range ordered structures.

### 4.7. Confocal Laser Scanning Microscopy (CLSM)

The morphological distribution of starch in different mixture samples before and after hydrothermal treatment was observed with CLSM (Leica CM3050S, Leica Bioystems Nussloch Gmbh, Germany). The starch suspension or paste was dyed with fluorescein isothiocyanate (FITC) (20 μL at 2 mg/mL) on a glass slide; meanwhile, it was washed three times with deionized water to remove the unbonded fluorescent dye, then covered with a microscopic glass. The images were recorded using CLSM with an emission range of 500–525 nm.

### 4.8. Sodium Dodecyl Sulfate Polyacrylamide Gel Electrophoresis (SDS-PAGE)

SDS-PAGE was conducted in a vertical electrophoresis cell using 10% separating gel and 6% stacking gel. A 50 mg sample was mixed with 1.5 mL of extraction buffer (Tris-HCl, 0.125 M, pH 6, containing 2% SDS (*w*/*v*), 10% glycerol (*v*/*v*) and 0.01% bromophenol blue (*w*/*v*)) for 3 h at room temperature, and the reducing SDS-PAGE was performed using the extraction buffer contained 5% (*v*/*v*) β-mercaptoethanol (β-ME). After extraction and centrifugation (10,000× *g*, 4 °C, 20 min), the supernatant was heated for 5 min in boiling water and loaded after cooling. The supernatant (matched in protein content) was loaded for SDS-PAGE analysis. Grey scale measurements were carried out with Image J software to denote the electrophoresis bands.

### 4.9. Chemical Interactions

Chemical interactions between the protein molecules of the prepared sample were determined according to the method described by Gómez-Guillén et al. [34], with minor modification. For this purpose, 300 mg of prepared sample was added to 10 mL of 0.05 M NaCl (S1), 0.06 M NaCl (S2), 0.06 M NaCl + 1.5 M urea (S3), 0.06 M NaCl + 8 M urea (S4), and 0.06 M NaCl + 8 M urea + 10 mM dithiothreitol (DTT) (S5). The resulting solutions were mixed and homogenized for extraction for 1 h at room temperature, followed by centrifugation (10,000× *g*, 4 °C, 20 min). The chemical interactions were analyzed based on the protein content in different solutions, where ionic bonds were presented as the difference in the protein content between S1 and S2 solutions; similarly, the difference in the protein content between S2 and S3, S3 and S4, and S4 and S5 were used to estimate the contributions of hydrogen bond, hydrophobic interaction, and disulfide bond, respectively.

### 4.10. Molecular Docking

The Autodock vina program (version 1.5.6, Scripps Research Institute, San Diego, CA, USA) was used to study the binding information between two gluten proteins and starch. The protein amino acid sequences for HMW-GSs of 1Bx7 (GenBank: SCW25213.1) from Yangmai 16 and 1By9 (Accession: CAA43361.1) from Yangmai 22 were obtained from the protein data bank, and tertiary structures of HMW-GSs were predicted using I-TASSER online server, based on the multiple-threading alignments by LOMETS and iterative TASSER simulations [35]. An amylose fragment containing 6 α-(1→4)-D-glucose units was built by Carbohydrate Builder on glycam.org using the GLYCAM06j-1 forcefield [36]. All molecules were minimized before docking. The lowest energy mode of the docking conformations was visualized by PyMOL (v 2.0). The software Ligplot (v 2.2.4) was used to interaction visualization between starch and HMW-GS.

### 4.11. Statistics Analysis

All data were expressed as mean ± standard deviation (SD) of three replicates. Data were analyzed using one-way analysis of variance (ANOVA), and Duncan’s multiple range test was used to compare the means with a significance using an SPSS package (version 10.0, SPSS Inc., Chicago, IL, USA). The probability value of *p* < 0.05 was considered significant.

## 5. Conclusions

The present study has shown that the disruptive effects of starch and aggregation behaviors of protein caused by heat processing could be accompanied by interactions with each other during heating. The multi-scale structural and physicochemical properties of starch during heating were significantly affected by different gluten proteins. More degradation of amylopectin and disruption of long-range molecular order was detected in the Y16-S mixture during heating, which was due to less competitive hydration and more water mobility in the Y16-S mixture compared with the Y22-S mixture, as the starch gelatinization was influenced by water availability. Similarly, the polymerization behaviors of Y22 and Y16 in corresponding mixtures were significantly and differentially affected by starch due to steric hindrance, especially at higher temperature. Fewer amylopectin branching points and a higher degree of short-range order in Y16-S allowed for more gluten–starch interactions in this system than in Y22-S mixtures.

## Figures and Tables

**Figure 1 ijms-23-08523-f001:**
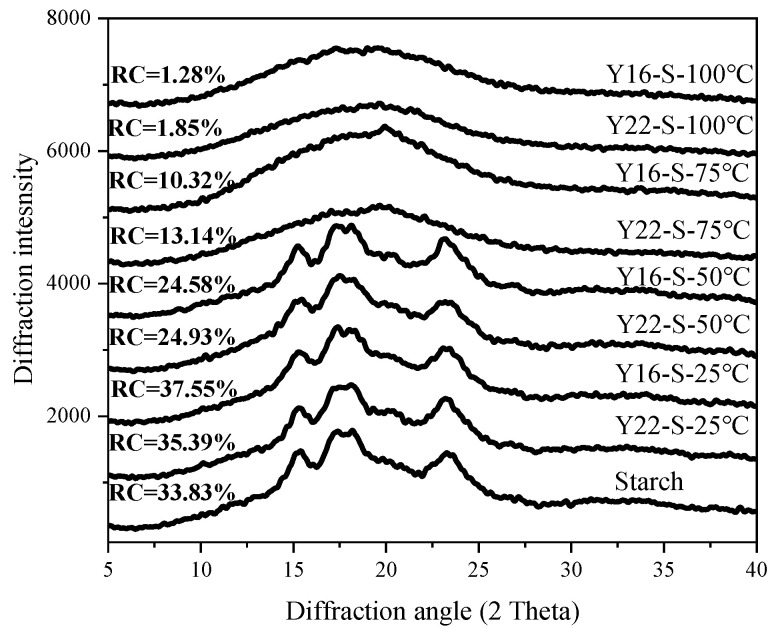
XRD pattern of native starch and different starch–gluten mixtures during heating (RC, relative crystallinity).

**Figure 2 ijms-23-08523-f002:**
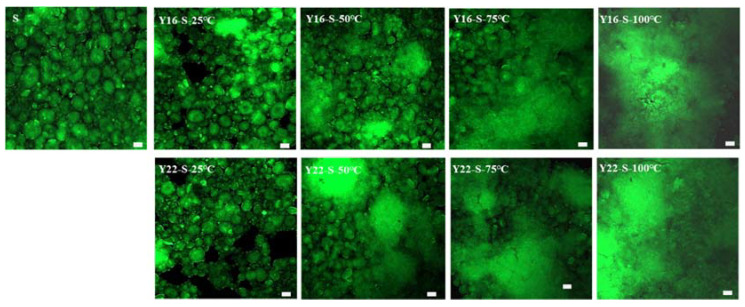
The CLSM of native starch and different gluten–starch mixtures during heating. Size bar = 50 μm.

**Figure 3 ijms-23-08523-f003:**
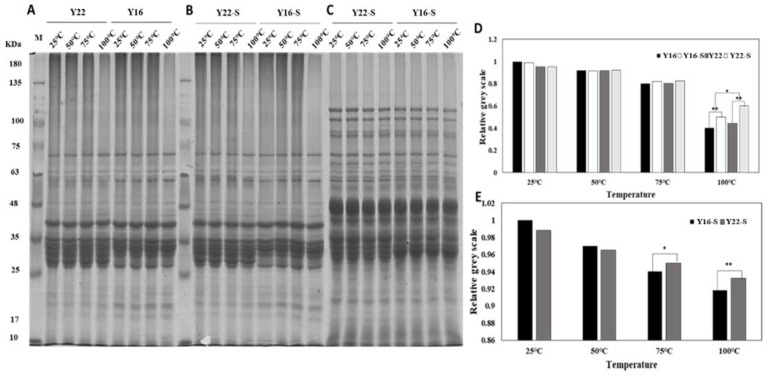
SDS-PAGE patterns and grey-scale measurements in different glutens and gluten–starch mixtures during heating. (**A**,**B**) Non-reducing SDS-PAGE patterns of Y22 and Y16, Y22-S and Y16-S mixtures during heating, respectively; (**C**) reducing SDS-PAGE pattern of Y22-S and Y16-S mixture during heating; (**D**) relative grey-scale measurements for SDS-soluble protein (130–200 KDa) from (**A**,**B**); (**E**) relative grey-scale measurements for HMW-GS from (**C**). * and ** indicate significant level of 5% and 1%, respectively.

**Figure 4 ijms-23-08523-f004:**
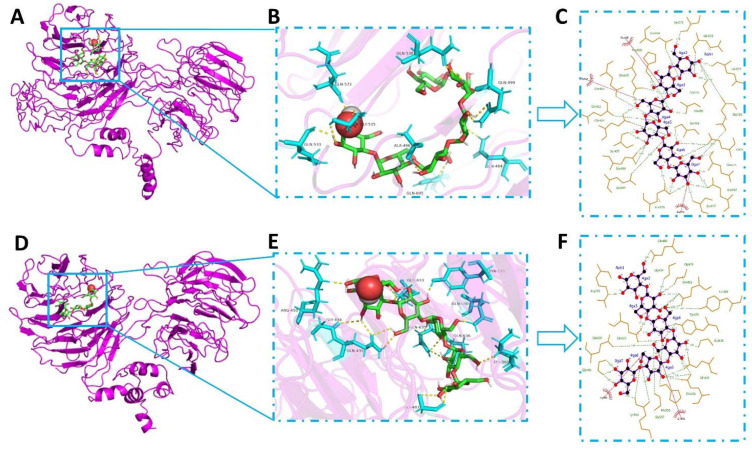
Molecular docking of HMW-GSs (Y16 and Y22) with starch. (**A**,**D**) represent the docking conformations of starch with HMW-GSs from Y16 and Y22, respectively. (**B**,**E**) represent the non-bonding interaction of corresponding docking conformations, and (**C**,**F**) represent the 2-D interaction of corresponding docking conformations.

**Table 1 ijms-23-08523-t001:** Swell power, solubility, pasting, and thermal properties of native starch and different gluten–starch mixtures during heating.

Sample	SP (g/g)	SI (%)	PV (cP)	TV (cP)	BV (cP)	FV (cP)	SV (cP)	PT (°C)	To (°C)	Tp (°C)	Tc (°C)	ΔH (J/g)
S	27.40 ± 0.02 ^a^	25.03 ± 0.03 ^a^	1903 ± 2 ^a^	1210 ± 13 ^b^	693 ± 6 ^a^	2227 ± 18 ^a^	1017 ± 5 ^a^	76.2 ± 0.03 ^f^	55.30 ± 0.37 ^g^	61.38 ± 0.22 ^g^	65.71 ± 0.34 ^g^	9.91 ± 0.16 ^a^
Y22-S-25 °C	17.64 ± 0.00 ^d^	18.18 ± 0.00 ^d^	1525 ± 3 ^d^	1244 ± 39 ^b^	282 ± 4 ^d^	1529 ± 16 ^e^	301 ± 1 ^d^	82.3 ± 0.17 ^d^	56.80 ± 0.23 ^f^	62.10 ± 0.53 ^f^	67.60 ± 0.32 ^f^	7.30 ± 0.12 ^b^
Y22-S-50 °C	18.22 ± 0.00 ^b^	22.73 ± 0.39 ^b^	1743 ± 12 ^b^	1377 ± 3 ^a^	367 ± 9 ^b^	1683 ± 10 ^b^	306 ± 6 ^d^	83.6 ± 0.95 ^c^	60.10 ± 0.53 ^d^	65.80 ± 0.33 ^d^	69.40 ± 0.32 ^d^	6.80 ± 0.32 ^c^
Y22-S-75 °C	13.22 ± 0.23 ^e^	17.05 ± 0.19 ^e^	992 ± 25 ^g^	886 ± 49 ^d^	76 ± 4 ^f^	1028 ± 45 ^h^	133 ± 9 ^e^	91.3 ± 0.00 ^b^	64.70 ± 0.65 ^b^	69.30 ± 0.55 ^b^	74.40 ± 0.22 ^b^	2.10 ± 0.22 ^e^
Y22-S-100 °C	12.3 ± 0.13 ^f^	14.72 ± 0.12 ^f^	1050 ± 28 ^f^	808 ± 24 ^d^	243 ± 3 ^e^	955 ± 6 ^i^	147 ± 18 ^e^	95.0 ± 0.03 ^a^	-	-	-	-
Y16-S-25 °C	17.92 ± 0.03 ^c^	18.18 ± 0.00 ^d^	1541 ± 2 ^cd^	1197 ± 8 ^b^	345 ± 2 ^c^	1595 ± 7 ^d^	405 ± 7 ^b^	78.4 ± 0.28 ^e^	57.90 ± 0.32 ^e^	64.30 ± 0.22 ^e^	68.50 ± 0.23 ^e^	6.80 ± 0.33 ^c^
Y16-S-50 °C	18.27 ± 0.04 ^b^	19.32 ± 0.19 ^c^	1587 ± 13 ^c^	1221 ± 17 ^b^	367 ± 16.26 ^b^	1638 ± 14.14 ^c^	417 ± 16.97 ^b^	78.6 ± 0.49 ^e^	61.60 ± 0.12 ^c^	68.70 ± 0.32 ^c^	70.10 ± 0.53 ^c^	5.30 ± 0.13 ^d^
Y16-S-75 °C	11.95 ± 0.00 ^g^	14.77 ± 0.19 ^f^	908 ± 12 ^h^	816 ± 16 ^d^	85 ± 8.48 ^f^	1113 ± 12.02 ^g^	297 ± 14.14 ^d^	91.7 ± 0.03 ^b^	66.40 ± 0.22 ^a^	72.90 ± 0.12 ^a^	74.70 ± 0.55 ^a^	1.70 ± 0.25 ^f^
Y16-S-100 °C	11.88 ± 0.01 ^g^	14.32 ± 0.26 ^f^	1188 ± 17 ^e^	1037 ± 75 ^c^	251 ± 5.65 ^e^	1451 ± 23.33 ^f^	365 ± 19.09 ^c^	74.0 ± 0.81 ^g^	-	-	-	-

Results are presented as means ± SD. Values in the same column with different letters indicate significant differences (*p* < 0.05). S, starch suspension prepared with 3.0 g (dry basis) of starch and 25 mL of distilled water. Y22 and Y16, glutens from weak (Yangmai 22) and medium (Yangmai 16) gluten wheats. The suspensions prepared with Y16 (0.5 g) and S (3 g) mixture with 25 mL distilled water and heated at 25, 50, 75, and 100 °C were designated as Y16-S-25 °C/50 °C/75 °C/100 °C, Y22 (0.5 g) and S (3 g) mixtures at 25, 50, 75 and 100 °C were designated as Y22-S-25 °C/50 °C/75 °C/100 °C, respectively. SP, swelling power; SI, solubility; PV, peak viscosity; TV, trough viscosity; BV, breakdown viscosity; FV, final viscosity; SV, setback viscosity; PT, pasting temperature. To, Tp, Tc, and ΔH represent the onset temperature, peak temperature, conclusion temperature, and enthalpy.

**Table 2 ijms-23-08523-t002:** Short-range molecular orders of starch in native starch and different gluten–starch mixtures during heating determined by ATR-FTIR and Raman spectra.

Sample	IR Ratio of 1047/1022 cm^−1^	IR Ratio of 1022/995 cm^−1^	FWHM at 480 cm^−1^
S	0.61 ± 0.003 ^a^	1.00 ± 0.021 ^g^	16.54 ± 0.11 ^d^
Y22-S-25 °C	0.44 ± 0.004 ^f^	2.10 ± 0.089 ^d^	16.31 ± 0.27 ^e^
Y22-S-50 °C	0.44 ± 0.002 ^f^	2.23 ± 0.031 ^c^	17.06 ± 0.05 ^c^
Y22-S-75 °C	0.40 ± 0.008 ^g^	3.60 ± 0.080 ^b^	17.32 ± 0.08 ^b^
Y22-S-100 °C	0.36 ± 0.003 ^h^	4.86 ± 0.165 ^a^	17.74 ± 0.05 ^a^
Y16-S-25 °C	0.62 ± 0.003 ^a^	0.87 ± 0.018 ^h^	16.14 ± 0.13 ^ef^
Y16-S-50 °C	0.57 ± 0.002 ^b^	1.08 ± 0.015 ^g^	15.91 ± 0.12 ^f^
Y16-S-75 °C	0.52 ± 0.002 ^c^	1.54 ± 0.011 ^ef^	16.73 ± 0.07 ^d^
Y16-S-100 °C	0.49 ± 0.001 ^e^	1.45 ± 0.017 ^f^	17.54 ± 0.15 ^ab^

Results are presented as means ± SD. Values in the same column with different letters indicate significant differences (*p* < 0.05). FWHM, full width at half maximum.

**Table 3 ijms-23-08523-t003:** Chemical interaction changes in different gluten–starch mixtures during heating.

Sample	Ionic Bonds (g/L)	Hydrogen Bonds (g/L)	Hydrophobic Bonds (g/L)	Disulfide Bonds (g/L)
Y22-S-25 °C	0.52 ± 0.03 ^a^	3.46 ± 0.07 ^d^	26.24 ± 0.01 ^d^	25.33 ± 1.00 ^f^
Y22-S-50 °C	0.48 ± 0.01 ^a^	3.67 ± 0.01 ^c^	26.76 ± 0.46 ^c^	28.01 ± 0.03 ^e^
Y22-S-75 °C	0.46 ± 0.00 ^a^	3.42 ± 0.00 ^d^	23.76 ± 0.01 ^f^	31.56 ± 0.06 ^d^
Y22-S-100 °C	0.29 ± 0.03 ^b^	3.27 ± 0.03 ^e^	16.69 ± 0.07 ^h^	38.85 ± 0.04 ^b^
Y16-S-25 °C	0.48 ± 0.01 ^a^	4.34 ± 0.03 ^a^	28.83 ± 0.11 ^a^	31.25 ± 0.28 ^d^
Y16-S-50 °C	0.31 ± 0.03 ^b^	4.26 ± 0.01 ^a^	27.98 ± 0.37 ^b^	30.89 ± 0.89 ^d^
Y16-S-75 °C	0.21 ± 0.02 ^b^	3.86 ± 0.01 ^b^	25.27 ± 0.03 ^e^	36.45 ± 1.12 ^c^
Y16-S-100 °C	0.19 ± 0.07 ^c^	3.75 ± 0.06 ^c^	18.70 ± 0.00 ^g^	52.96 ± 0.32 ^a^

Results are presented as means ± SD. Values in the same column with different letters indicate significant differences (*p* < 0.05).

## Data Availability

Not applicable.

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
