# Peer review of "Effects of Different Gluten Proteins on Starch’s Structural and Physicochemical Properties during Heating and Their Molecular Interactions"

_ijms, 2022, doi:10.3390/ijms23158523_

Round 1

Reviewer 1 Report

Table 1 is not very readible. Please precise amount of starch in the disspersion (in Table 1 description) Line 117-125: all abbrev. should be as bottom Table captions.

XRD analysis: the results are disscused quite poorly, eg. what is a source of V-type strucure formation after processing?

Reviewer 2 Report

I think that this is a technically sound work that yielded interesting results. However, the formatting and the writing need to be substantially improved, so that the work is presented in the best possible way. 

A general questions from me relates to the fact that it has been shown that B type starch granules can affect gluten network development differently than A-type granules. Was anything known about the granule distribution in the sample, and is this a point that could be added somewhere?

A general comment: Why is "wheat" abbreviated with "G" (L13/14)? I assume the authors are actually referring to gluten with "G", but the way the sentence is written makes it seem as if it is wheat that they are abbreviating.

L17: The authors write "especially SG" - should it not be "especially in SG"?

L17/18: "These led to the less degradation of amylopectin branch and long-range molecular order of starch in SG and starch (SG-S) mixture during heating than those in HG and starch (HG-S) mixture" - do the authors mean "amylopectin branches"? The way the sentence is written now makes it seem as if it would be "branch order", which is not a term I have heard before - or it would be "amylopectin branch", which would be grammatically incorrect. Also, "and starch (SG-S) mixture" is unclear (grammatically). I think the authors mean that this occurred more in gluten from soft wheat as well as soft wheat gluten-starch mixtures than in hard wheat gluten and its blends with starch. Please rephrase the sentence to make it clearer. 

L20: " polymerization behaviors of two glutens" - do the authors mean that the polymerization of gluten proteins from both soft and hard wheat was influenced? The phrasing is currently a bit strange - also, if the authors are referring to SG and HG, then should "gluten" not be abbreviated with G here? 

L21: " branch point" - should this not be in the plural form?

L24: "steric hindrance in starch-gluten network" - either write "in the" or use the plural form, "starch-gluten networks" (preferably the latter).

L24: "the contribution of more residual short-range ordered starch and hydrogen bonds in HG-S mixture during heating" - the grammar is off here, please revise.

L51: Should it say "meat [6] and soy [7]"? Also, "had elucidated" is not the right phrase here

L52: "Meanwhile, the gluten protein forms" - meanwhile is not the right term; and there are several proteins that form gluten, it is not just one, so use the plural form (also edit L53 accordingly). Actually, I'd write "proteins form the gluten network"

L56-61: Check the grammar of this sentence, in particular subject-verb agreements

L82-85: The authors wrote: "Therefore, this study will use the gradual heating process, which involves the transformation and interaction between the transient and continuous states of starch and gluten. Meanwhile, glutens from two different wheat flours were taken into account to differentiate the main interactions between starch and different gluten". These were just representative flours from soft and hard wheat. In addition, I am not sure if the term "meanwhile" is the correct conjunction here (there were a few more places in the manuscript where I thought it was not the right kind of conjunction). 

L115: Are all decimals necessary? Could the number of significant figures be reduced? Moreover, would the values not be much easier to read if the letters denoting statistical significance were written in superscript? 

L127-128: The word "decreased" is written twice ("both decreased significantly decreased")

L142 - 143: It was not entirely clear to me if a relative crystallinity of 1.85 vs 1.28 would matter (the authors emphasize that the crystallinity at 100 deg C differed (when the starch was essentially gelatinized) was higher in the system with gluten from soft wheat. A few lines below, in L174-175, the authors also write how the granule morphology was altered after heating to 100 deg C. The standard deviation is also not stated, so it is not clear if the relative crystallinities were significantly different at such a high temperature.

L147: "short-range ordered" sounds grammatically incorrect. It is fine when used together with "structure", like in the title for section 2.3 (L133), but by itself it sounds wrong.

L147-161: Some explanation should be added about what these ratios actually denote

L159: "in both two mixtures" - I think "two" is redundant

L167: Use past tense for results, and check for subject-verb agreement

L170: Check the grammar in the sentence

L173: Subject-verb disagreement

L174: "especially more obviously" is not correct

L180: I am more familiar with SDS-PAGE conditions being described as reducing or non-reducing (instead of writing 'reduced'/'non-reduced'). Also, replace "to analysis" with "to analyze". In L181, I don't think the word "behavior" is needed. 

L183: The authors are using the term 'soluble' - but that depends on what solvent is considered. Would it not be better to write "solubilized", or maybe specifically refer to them as "SDS-soluble" here? In L188, is the term "soluble" necessary? The authors are referring to SDS-PAGE results obtained either with or without beta-mercaptoethanol. If the proteins had not been solublized, they could not have been analyzed by SDS-PAGE. So it's just part of the method. It does not mean that they were soluble in the dough. I am not sure why the authors emphasize the solubility so much here? The extraction system is different than used for section 2.6 (because the purpose was different there), so to me it's just confusing.

L190: Is the word "relative" needed here? I am not sure why the authors included it. The phrase "especially more obviously" sounds incorrect.

L191-192: Can the grey scale itself be "significantly higher"? Would it not be the intensity that is higher?

L196: I would write "band density", like in L199, not "bands density"

Figure 3: I was surprised that the bands for the two gluten systems were so similar. I would have expected more variability between gluten from soft and hard wheat, especially in the HMW region. In L337, the names of the two flours are given and I've noticed that they are both called Yangmai and just differ in their number. Are these varieties closely related? 

L209 and section 2.6: The original reference cited by the authors only listed the results as protein solubility and then specified the reagents in the table. The authors of the current manuscript selected "Chemical force interaction changes" as the descriptor in their title. But based on such a title, I would expect a parameter with different units. A force is measured in N, but what the authors are reporting is a solubility (or changes therein, depending on changes in the solvent). I would rephrase the title and reconsider how they describe the results in section 2.3, i.e. what terminology to use. I think the term "chemical interactions" is more common. Is the term "force" really needed (this applies to the whole manuscript)? I am also more used to the term "hydrophobic interactions" instead of "hydrophobic bonds", and the authors have used this term (i.e., hydrophobic interactions) in L420 as well. 

Table 3: The authors included starch, but not SG or HG in this table - why? Also, where is the value for HG-S heated to 100 deg C?

L225: I believe it is called "in silico", not "in silicon"

L240: "theoretical speculated" does not sound good to me. How about "hypothesized"?

L241: In flour systems, several proteins have been shown to be associated with the starch granule surface. Would the authors say that gliadins surround the granules in model systems more than in flour? 

L241/242: It is not clear to me if the authors mean to say that these other studies have found results to corroborate this? If so, which sources, 15 and 16 (they should be cited again if this is the case)? What is meant by "variation in gluten protein"? Differences in the ratio of glutenins to gliadins?

L243: "the" before "purified" can be removed

L244: "which was to be devoted" sounds a bit odd; "systematically" is not the right form of the word here

L248-252: This was not clear to me (also, the grammar seems off): If there are more water molecules of higher mobility in the HG system than in the SG system, then why would this water not be available for the starch to gelatinize? 

L252-255: This sentence was also unclear to me. Are the authors saying that gliadins form double helices with amylopectins? I have not heard of that. And then why would the gliadins be spherical, if they were indeed part of a double helix? I checked reference [17], and could not find anything in there that would clarify this sentence.  

L265-268: In reference [12], it was observed that a higher glutenin to gliadin ratios increased peak and trough viscosity as well. To me, this seems to be in line with the results here regarding the effect of HG and SG, and I would suggest citing it. As a general comment, I think the authors could highlight a bit more which of their results are in agreement with other literature and which ones perhaps aren't.   

L284: This sentence was a bit unclear - what is "closer formation"?

L285/286: The authors have put the term complexation in quotation marks, could it be clarified why?

L289: For the most part, starch and gluten-forming proteins are described as thermodynamically incompatible, which is also mentioned e.g., by reference [23] the authors are citing (and the authors also mention it a few lines below in L301 and cite [25]). In the authors view', how does this fit with their results? 

L289: I think the word "that" should be removed. 

L291: I would not say that chemical groups are "released" by gluten. Do the authors mean that they become accessible due to unfolding?

L305/306: What about differences in the gluten composition between SG and HG - how much would that influence the polymerization in comparison to the effect that starch would exert?

L309: To me, "less" would sound better than "lower"

L310: In general, I think that "conducive" would be the better word choice than "conductive" to describe the situation. But the sentence is quite complicated in its structure. Which of the components is the active part, is it the amylopectin that it disrupting the interactions? I am asking because "lower degradation" cannot in itself "disrupt", doesn't there have to be a component that does the disrupting? If so, that would be more than being conducive. Please consider simplifying the sentence (also, "as mentioned above" may not be necessary - just because the sentence is already long, I'd cut it).

L313/314: "the presence of branch points (...) contribute to steric hindrance in suppressing the structural changes". I think that the authors mean to say that more branching points lead to more steric hindrance, which hinders unfolding and aggregation. The structural changes that the sentence refers to seem to be the ones in glutenins, that would allow GMP formation. The way the sentence is currently written is quite complicated. The same is true for the next sentence:

L314/316: It was also not so easy to understand. Essentially, it seems to me as if the authors are suggesting that GMP formation in the SG-S system was influenced more by amylopectin than in the HG-system. The reason why amylopectin could influence the system is because of steric hindrance, a consequence of its large molecular weight and abundance of branching points. However, this does not fully explain the situation. The same starch, with the same amylopectin, was used. This seems to be the reason why in the next sentence: 

L317-319, the authors write that the more chemical interactions in HG-S mitigated the hindering effect of amylopectin. Table 3 reports the changes in protein solubility when the solvent was modified. It does not report the actual values, which is something I would suggest to put into supplementary data. Overall, this section of the paper requires more clarification, I think it is one of the most crucial segments. The authors are arguing that the proteins in gluten from hard wheat exhibit more interactions, and therefore amylopectin cannot disrupt it as much as it does in the system from soft wheat. But is steric hindrance (due to the branching points) necessarily suppressed by non-covalent interactions (which are quite easy to break)? Even though there are some significant differences among SG-S and HG-S in the values for ionic and hydrogen bonds, as well as for hydrophobic interactions, are these differences really expected to have a crucial impact? A difference of e.g., 3.27 vs 3.75 for H bonds at 75 deg C may be significant, but how much would such a small difference matter? The one value that strikes me as quite different though is the change in solubility based on disulfide bonds for systems heated to 75 deg (it would be nice to know what happened at 100 deg C to HG-S...which, however, is missing from the table). A larger contribution of disulfide bonds is in line with expectations for gluten from hard wheat. It would thus fit with the information provided in 4.1., since the HG system contained more glutenins, especially more HMW-GS, than SG. If there are S-S groups, then would this promote the non-covalent interactions too?  

L337: The Latin name for wheat should be written in italic font, except for the L. 

L345: Usually, author's first names are not part of in-text citations. 

L347: The protein contents of the 2 gluten samples were slightly different -  are the values given in L347 on a dry or wet basis? After the gluten and starch samples were mixed, they were lyophilized by the authors. Not all moisture is removed in this process - was the residual moisture content measured (it could have been slightly different among the samples)? For SDS-PAGE, the authors used 50 mg (L407), but were the samples matched in protein content before loading onto the gels?

L351: "was added with" is incorrect English. It should be "added to" or "combined with". Also, the abbreviations have been defined already, why repeat it here?

L354: The authors write that the samples were heated at these respective temperatures. However, would it not take a while before the higher (especially 100 degrees) temperatures are reached, and was this assessed?

L356: How were the samples milled?

L363: Should the instrument (RVA) not be introduced the first time it is mentioned in the materials and methods section, which would be the section above?

L390: Should there really be a comma after "et al."? 

L439: Why is the font size all of a sudden different?

L440: I think that the word "that" needs to be inserted after "shown", otherwise the sentence does not sound grammatically correct to me.

L449-450: Please revise the sentence, the grammar is off

L507: I believe the author's name is BeMiller, not Bemiller. 

Round 2

Reviewer 2 Report

The authors have revised their manuscript and responded to reviewer comments. However, there are still many issues with the writing, and I think these need to be addressed before the manuscript can be published. It is important that the text is grammatically correct and that it is understandable for readers.

L13/14: "The effects of glutens from two different wheat flours (a weak-gluten wheat gluten (Yangmai 22, Y22) and a medium-gluten wheat gluten (Yangmai 16, Y16)) on starch (S) structural and physicochemical properties during heating and their molecular interactions were investigated in this study." It would sound better to write "(...) two wheat flours (a weak gluten (Yangmai 22, Y22) and a medium-strong gluten (Yangmai 16, Y16)) ..." because I think the authors mean to say "medium-strong" or "of medium strength".

L17: "due to competitive hydration effect" - remove the word effect, "competitive hydration" by itself is sufficient and also sounds better

L18: "These led to the less degradation of long-range molecular order of starch and amylopectin branch in Y22-starch mixture (Y22-S) (...)" - this sounds really complicated. Why not write "Thus, over heating the long-range molecular order and amylopectin branches of starch were better preserved in the Y22-starch mixture (Y22-S) than in the Y16-starch mixture (Y16-S)."

L21: " of glutens from two wheat varieties" - why not write "of both glutens"? It has already been mentioned that two wheat varieties were used, so it does not need to be written here again (it just makes the sentence longer and more complicated, and it almost sounds as if the authors are now talking about two other varieties).

L21-23: "Residual amylopectin branch points in gelatinized starch during heating acted as steric hindrance in starch-gluten networks, resulting the hindrance the aggregation and cross-linking of different gluten proteins at higher temperature." This sentence is long-winded and there are grammar issue, e.g., "resulting the hindrance" is not grammatically correct. As a general comment, there are many sentences in the manuscript that are difficult to read because there are repetitive statements in the same sentence. Here, it's "during heating" and "at higher temperature", and "steric hindrance" and "hindrance". I suggest writing something like "During heating, residual amylopectin branching points restricted the aggregation and cross-linking of gluten proteins due to steric hindrance".

L50: Remove "the" before "granule swelling"

L51-53: "Furthermore, the use of the purified starch and protein from meat [6] and soy [7] as model materials had shown some interactions of individual macromolecules." - I suggest replacing it with "Furthermore, in models using purified starch and proteins from meat [6] as well as soy [7] interactions among these macromolecules have been shown".

L57-58/58-61: Both sentences are grammatically incorrect. For example, one cannot write "since" in a sentence and then not write the consequence in the same sentence. There are also repetitive statements again. I suggest this alternative:

"However, since previous studies rarely focused on interactions between starch and gluten during heating, little is known about specific interaction mechanism in such model systems, for example how residual short-range order in gelatinized starch influences gluten polymerization."

L82: "Therefore, this study will use the gradual heating process (...)" - change this to "Therefore, this study used a gradual heating process (...)"

L84-86: "Glutens of two different representative flours from weak and medium gluten wheat were taken into account to differentiate the main interactions between starch and different gluten." I feel that this is another example of a sentence with repetitive statements. Moreover, in their response to one of my reviewer comments (point 11), the authors provided information that could be interesting and informative to readers. So would the authors say that the wheat flours have the same allelic profile for HMW-GS? If so, why not incorporate that? It actually is a strength of the study that two wheat varieties were used that had such similar protein profiles. For example: "Gluten from two wheat varieties with the same high-molecular weight (HMW) glutenin profile but of different strength were used." Or maybe "Gluten from two wheat relatives with the same allelic variation of high-molecular weight (HMW) glutenins but of different gluten strength were used" - or something similar. I don't think that the names of the parents necessarily need to be mentioned, because most readers will not be familiar with them (so it will not make the text more understandable). But I would clearly state what is similar and what is different about them - why exactly are they different, by the way? Usually, the HMW-GS are mostly responsible for the strength. It seems to me that whoever designed the study had a clear rationale for specifically using these wheat varieties. Currently, the text does not make that fully clear, but I suspect that at least partly the gliadin to glutenin ratio that the authors mention elsewhere is responsible. My point is that based on the information in the response letter, it seems that these weren't just two random wheat varieties that were chosen. Maybe the information is scattered throughout the paper. But it would make a much more convincing case for the study design if the authors summarized it early on in the paper.

L142/143: "suggesting that the formation of lipid-starch complex at higher temperature due to small amount of lipid in native starch " does not sound grammatically correct

L153-162: I looked up reference [32], a previous study by these authors which is cited as reference for the FT-IR method. I think that it in [32] it is much clearer stated what these ratios mean, and I therefore think that the sentence here should be revised. Is it really the "structure" of "short-range order" or is it more the extent or the degree of it? Another (minor) issue is that there are a few samples where the ratios are not significantly different to the native starch sample, so technically the statement that in L156/157 is not fully accurate. Here is a suggestion for how to phrase these sentences; the authors can modify them as they see fit: "FT-IR and FT-Raman spectroscopy were performed to give information on the short-range order of starch. The absorbance ratio of 1047/1022 cm-1 is indicative of the degree of order in the outer regions of starch while the ratio of 1022/995 cm-1 denotes proportions of amorphous versus ordered structures. As shown in Table 2, native starch had a significantly higher ratio of 1047/1022 cm-1 and significantly lower ratio of 1022/995 cm-1 than all Y22-S samples. In these samples, the 1047/1022 cm-1 and 1022/995 cm-1 ratios progressively decreased and increased as soon as samples were heated. In contrast, the ratios were only significantly different between Y16-S and native starch after samples had been heated to 50 °C (1047/1022 cm-1) or 75 °C (1022/995 cm-1). Moreover, while the 1047/1022 cm-1 ratio decreased and the 1022/995 cm-1 ratio also increased in Y16-S over heating, the ratios were at all temperatures higher and lower, respectively, than in Y22-S samples."

L163: The authors wrote that the FWHM "was calculated to characterize the short-range ordered structure". To a reader who is not familiar with starch spectroscopy it is not clear what is different about this ratio than the 1047/1022 and 1022/995 ratios (I think it is the fact that Raman was used for it, but this is not stated anywhere). In fact, in the materials and methods, the authors write about FT-IR first and state that "the short-range ordered structure of starch sample was determined", so that is almost the same sentence as used in L162-164 for FWHM. The FWHM parameter is not explicitly listed in section 4.6.

L164: "and higher the FWHM at 480 cm-1 represents lower short-range ordered structures" does not sound grammatically correct. Maybe "and higher values for FWHM at 480 cm-1 represent fewer ordered structures due to short-range interactions" or "and higher FWHM values at 480 cm-1 represent a lower proportion of short-range ordered structures"?

L180: I would replace "cracked" by "disintegrated".

L190: "The non-reducing and reducing SDS-PAGE patterns" - remove "the" and "patterns" from the sentence

L191-192: I would remove "pattern at different heating temperature" from the sentence because it sounds a bit odd and the information is redundant

L198: I would split this very long sentence into two, right after "mixture" and then begin the next sentence with "Although" instead of "though"

L200:  "more obvious" is maybe not the best term (it sounds too casual), how about "more pronounced" or "more intense"?

L219: I do not think that "The contents of chemical interactions in two mixtures during heating are shown" is a correct statement. Chemical interactions do not have 'amounts' or 'contents'. The authors measured the contents of proteins in various solvents that selectively dissociate proteins based on chemical interactions.

L234: I think that "in silico" should be italicized

L250: "the starch granule which is covered by gliadin" - this sounds as if there is only one granule in the whole system, but there are many granules

L250-252: " Therefore, the variation in gluten protein influenced the interaction" & "Taking into this account, purified starch and gluten" - this is still not clear from how it is written. Do the authors mean that references 15 and 16 came to the conclusion that differences in gluten composition influence the interactions with starch, or is this the interpretation of the authors of the current manuscript? Also, the word order for "into" and "this" needs to be switched.

L258: "this could further interpret the more decrease of swelling power" - this is not grammatically correct. It's also another really long and convoluted sentence that could easily be split into two.

L261: "hydrophilic polar" - one of these terms can be removed, they essentially mean the same thing. Also, it's not really clear why these groups "remain" on the surface? Do the authors mean that they remain accessible? Either way, "remain" may need to be changed to "remaining", in its current form the sentence is not grammatically correct

L262: Is it a true "link" or would it be better to write "interaction"?

L265: Replace "more" with "a higher"

L285-286/L287-292: The  authors are referring to table 3 in L286, but these ratios are listed in table 2. Moreover, to me some of the statements did not seem grammatically correct and then L287-292 are one very long and complicated sentence. In L291 the authors write "formation of amorphous and crystalline regions" - but the crystalline regions are not formed, they are already present in the native starch. They could just be retained. I think that this could all be simplified without losing critical information. I suggest writing something like the following: "These results were consistent with data shown in Table 2, which indicated that some degree of short-range order remained over heating. The change in short-range molecular order was influenced by the type of gluten present. At various temperatures, Y16-S had a higher absorbance ratio at 1047/1022 cm-1 and lower values for the 1022/995 cm-1 ratio as well as FWHM at 480 cm-1 than Y22-S, suggesting that more of the short-range ordered structure of starch remained in Y16-S over heating than in Y22-S."

L294-296: There are grammar issues and it is also confusing that Y16 would promote its own interaction. I suggest something like "Our data indicate that Y16 had stronger interactions with starch during heating compared to Y22, and this preserved the short-range order in starch to a larger extent".

L468-472: The sentence contains "during heating" twice. Is that necessary?

L474: I think "due to the contribution of steric hindrance of starch" can be shortened to "due to steric hindrance". It would be easier to read that way, and the crucial information is still preserved.

L475-477: "During heating, compared with Y22-S mixture, less amylopectin branch point and more residual short-range molecular order structure in Y16-S mixture could mitigate the steric hindrance between starch and Y16, and cause more intense interaction between Y16 and starch." I had to read it several times because it was so complicated. If the authors think that it contains the same meaning, I would change it to "Fewer amylopectin branch points and a higher degree of short-range order in Y16-S allowed for more gluten-starch interactions in this system than in Y22-S mixtures".

L414 & 569-570: Is the reference cited correctly? Other authors who have cited this paper have listed Brandt as the first author and Chikishev as 2nd author.
